# Influence of Guided Tissue Regeneration Techniques on the Success Rate of Healing of Surgical Endodontic Treatment: A Systematic Review and Network Meta-Analysis

**DOI:** 10.3390/jcm11041062

**Published:** 2022-02-18

**Authors:** Álvaro Zubizarreta-Macho, Roberta Tosin, Fabio Tosin, Pilar Velasco Bohórquez, Lara San Hipólito Marín, José María Montiel-Company, Jesús Mena-Álvarez, Sofía Hernández Montero

**Affiliations:** 1Department of Implant Surgery, Faculty of Health Sciences, Alfonso X El Sabio University, 28691 Madrid, Spain; rtosi@myuax.com (R.T.); ftosi@myuax.com (F.T.); mvelaboh@uax.es (P.V.B.); lsanhmar@uax.es (L.S.H.M.); shernmon@uax.es (S.H.M.); 2Department of Surgery, Faculty of Medicine and Dentistry, University of Salamanca, 37008 Salamanca, Spain; 3Department of Endodontics, Faculty of Health Sciences, Alfonso X El Sabio University, 28691 Madrid, Spain; jmenaalv@uax.es; 4Department of Stomatology, Faculty of Medicine and Dentistry, University of Valencia, 46010 Valencia, Spain; jose.maria.montiel@uv.es

**Keywords:** endodontic surgery, periapical lesion, guided tissue regeneration, bone graft, membrane, platelet rich fibrin

## Abstract

Several regeneration techniques and materials have been proposed for the healing of bone defects after surgical endodontic treatment; however, the existing literature does not provide evidence on the most recommended techniques or materials. The aim of the present systematic review and network meta-analysis (NMA) is to summarize the clinical evidence on the efficacy of guided tissue regeneration techniques (GRTs). The PRISMA recommendations were followed. Four databases were searched up to December 2021. Randomized clinical trials (RCTs) with a minimum follow-up of 6 months were included. The risk of bias was assessed using the Cochrane Collaboration tool. A fixed effects model and frequentist approach were used in the NMA. Direct GRT technique comparisons were combined to estimate indirect comparisons, and the estimated effect size of the comparisons was analyzed using the odds ratio (OR). Inconsistency was assessed with the Q test, with a significance level of *p* < 0.01, and a net heat plot. A total of 274 articles was identified, and 11 RCTs (6 direct comparisons of 15 techniques) were included in the NMA, which examined 6 GRT techniques: control, Os, PL, MB, MB + Os, and MB + PL. The MB + Os group compared to the control (OR = 3.67, 95% CI: 1.36–9.90) and to the MB group (OR = 3.47, 95% CI: 1.07–11.3) showed statistically significant ORs (*p* ˂ 0.05). The MB + Os group presented the highest degree of certainly (P-score = 0.93).

## 1. Introduction

Bacterial infection plays an important role in establishing pulp tissue inflammation, which may lead to subsequent pulp necrosis and the formation of periapical lesions [1]. The complete removal of, or at least significant reduction in, the bacterial load during nonsurgical endodontic treatment is an important factor determining the final prognosis of root canal treatment. However, the development of apical periodontitis was reported in 44.9% of studied cases [2], mainly related to persistent or secondary endodontic infections [3].

Endodontic surgery is recommended after unsuccessful retreatment, when retreatment is impossible, or when there is an unfavorable prognosis [4]. Surgical endodontic procedures include removing necrotic and infected periapical tissues, resecting the apical part of the tooth (apicoectomy), and preparing the root-end cavity for the insertion of retrograde filling material [5]. Conventional endodontic surgery has been reported to result in a complete periapical tissue healing rate of 90% [6].

Recently, guided tissue regeneration (GTR) techniques have been widely used in medicine, including in dentistry, to improve tissue healing. Furthermore, GTR techniques have been recommended as an adjunct to endodontic surgery to promote periapical tissue healing and improve the treatment outcome [7].

Complete periapical healing involves the regeneration of alveolar bone, periodontal ligament cells, and cementum [8]. However, the surrounding connective tissues may grow into the osseous defect, preventing bone healing [6]. GTR techniques have been proposed as an adjunct to endodontic surgery approaches to promote bone healing and prevent the collapse of connective tissues [9].

Numerous studies reported the clinical effectiveness of GTR techniques to promote healing and improve the outcome of surgical endodontic treatments [10,11]. However, the wide range of available biomaterials, the treatment protocols, and the lack of standardization in assessment criteria lead to inconsistent and confusing results. Therefore, an evidence-based review and meta-analysis of the available literature regarding the influence of GTR techniques on the outcome of surgical endodontic treatment is necessary to help clinicians select the most predictable tissue regeneration technique for surgical endodontic treatment success.

Network meta-analysis (NMA) extends the principles of meta-analysis to the evaluation of several treatments in a single analysis, comparing multiple treatments simultaneously by combining direct and indirect evidence within an array of randomized controlled trials [12]. It is the best tool to examine the success rates of different procedures, such as GTR techniques in endodontic surgery.

The aim of the present study is to conduct a systematic review and NMA to analyze the influence of GTR techniques on the success rate of surgical endodontic treatment. The null hypothesis (H0) was that GTR techniques do not influence the success rate of surgical endodontic treatment.

## 2. Materials and Methods

### 2.1. Study Design and Registration

This systematic review and NMA was conducted following the Preferred Reporting Items for Systemic Reviews and Meta-Analyses (PRISMA, http://www.prisma-statement.org, accessed on 30 July 2020) guidelines. The review also fulfilled the PRISMA 2009 Checklist [13]. The registration number is CRD42020203447 (PROSPERO).

### 2.2. Literature Search Process

The search strategy was based on the following population, intervention, comparison, outcome (PICO) question: in adult patients undergoing endodontic surgery (P), does the use of regeneration techniques (I) compared to not applying regeneration techniques (C) influence the success rate (O)? An electronic search was conducted in the PubMed, Scopus, EMBASE, and Web of Science databases. The search covered all of the literature published internationally up to December 2021. The search included the following medical subject heading (MeSH) terms: “apicoectomy”, “periapical surgery”, “endodontic surgery”, “periapical lesion”, “surgical endodontic treatment”, “root-end surgery”, “root-end resection”, “periradicular surgery”, “guided tissue regeneration”, “bone graft”, “bone regeneration”, and “membrane”. The Boolean operators applied were OR and AND. The search terms were structured as follows: ((“apicoectomy”) OR (“periapical surgery”) OR (“endodontic surgery”) OR (“periapical lesion”) OR (“surgical endodontic treatment”) OR (“root-end surgery”) OR (“root-end resection”) OR (“periradicular surgery”)) AND ((“guided tissue regeneration”) OR (“bone graft”) OR (“bone regeneration”)) AND ((“membrane”)). Two researchers (R.T. and A.Z.M.) independently conducted the database searches in duplicate. Titles and abstracts were selected by applying the inclusion and exclusion criteria. One researcher (R.T.) extracted data for the relevant variables. The systematic review was carried out by R.T., and two researchers not involved in the selection process (A.Z.M. and J.M.C.) performed the subsequent meta-analysis.

### 2.3. Inclusion and Exclusion Criteria

The inclusion criteria for the selected studies were as follows: randomized clinical trials (RCTs) that had a minimum follow-up period of at least 6 months; studies that analyzed GTR techniques (bone graft, membrane, membrane plus bone graft, platelet-rich plasma, or membrane plus platelet-rich plasma) or compared GTR techniques with a control treatment; patients that were 18 years old or older; and endodontic surgery procedures that were used to treat apical and/or apical-marginal lesions. No restrictions were placed on the year of publication or language.

The exclusion criteria for the selected studies were as follows: systematic or bibliographic reviews, clinical cases, case series, retrospective studies, and editorials and studies, including patients younger than 18 years.

### 2.4. Data Extraction

The following data were extracted from each study by independent reviewers (S.H.M. and J.M.A.): author and year of publication, title, journal in which the article was published, sample size (n), follow-up time, measurement procedure, type of GTR technique, success rate, periapical reduction, and bone density. The success of healing was analyzed according to the radiographic criteria established by Rud et al. [14] and Molven et al. [15], with complete healing defined as the reformation of periodontal space (intact lamina dura) with one cavity filled with bone (which can be of different radiopacity) and complete bone repair, but no discernable PDL around the apex. A third reviewer (P.V.B.) was consulted if the independent reviewers did not agree.

### 2.5. Risk of Bias

The risk of bias in the selected studies was assessed using the Cochrane Collaboration tool for methodological quality assessment of clinical trials [16]. This tool consists of 7 items that evaluate sequence generation, allocation concealment, participant blinding, assessment blinding, incomplete data, free selective reporting, and other sources of bias (Table 1 and Figure 1).

### 2.6. Data Synthesis and Statistical Analysis

The meta-analysis was carried out using a random effects model to estimate the success rate of endodontic surgery with and without GTR techniques, along with the confidence intervals. Heterogeneity among the combined studies for each treatment group was assessed using the I^2^ statistical index [17], which describes the percentage of total variation of studies due to heterogeneity and is not random. The effect of heterogeneity was quantified as being between 0 and 100% (low 0–25%, mild 25–50%, moderate 50–75%, high > 75%) [17]. The results of the meta-analysis are represented by forest plots.

Direct treatment comparisons were combined with a fixed effects model in a frequentist NMA to estimate indirect comparisons. The estimated effect size of the comparisons was analyzed by the OR. The inconsistency of studies included in the NMA was assessed with the Q test [18], with a significance level of *p* < 0.01, and a net heat plot [19].

Direct comparisons were performed using a NETWORK graph, and treatments were ranked on a scale of 0 to 1 using a P-score measuring the degree of certainty and indicating whether one treatment was superior to another [20].

Publication bias was analyzed using the trim and fill adjustment method for funnel plot asymmetry. In this analysis, each study was represented by a point, and the effect size and standard error were represented on the X-axis (logit transformed proportion). If there were no significant differences between the initial and adjusted estimates, the publication bias was considered to be low. R software was used with the Metaprop and Netmetaprop statistical packages to perform the meta-analysis.

## 3. Results

### 3.1. Results of the Search Process

The systematic electronic search identified 159 articles in PubMed, 40 in Web of Science, 64 in EMBASE, 12 in Scopus, and 1 in the gray literature, which was found in the bibliography of a previous review [21]. Of the 276 articles, 56 were discarded as duplicates using RefWorks (https://refworks.proquest.com/reference/upload/recent/, accessed on 14 August 2020). After reading the titles and abstracts, an additional 130 articles were eliminated, leaving 90 articles; a further 55 articles were rejected because they did not fulfil the inclusion criteria: they did not include complete success rate data, did not use in vivo patient data, or presented a minimum follow-up time of less than 6 months. Finally, 11 articles were included in the qualitative and quantitative synthesis because they included all of the required data and variables (Figure 2).

### 3.2. Qualitative Analysis

All 11 articles that were included were randomized clinical trials [22,23,24,25,26,27,28,29,30,31,32]. Among them, 7 studies analyzed both clinical and radiographic parameters [23,24,26,27,28,30,31], and 4 studies analyzed radiographic parameters, such as bone density and periapical defect volume [22,25,27,32]. Most of the studies presented a sample size of approximately 25–30 patients, although the sample size ranged from 25 [30] to 101 [23], with subject ages ranging from 18 to 70 years and a follow-up time from 12 to 24 months. The results are presented in Table 2.

### 3.3. Assessment of Risk of Bias

The methodological quality results were assessed using the Cochrane Collaboration tool and are shown in Table 1. All selected studies showed a low risk of bias related to incomplete data outcome assessment and other sources of bias. Moreover, most studies showed a low risk of bias related to sequence generation assessment and blinding of outcome assessors; however, most studies also showed an unclear risk of bias related to allocation concealment and participant blinding, and all studies showed an unclear risk of bias related to free selective reporting.

### 3.4. Quantitative Analysis Results

Odds ratios among regeneration techniques for the success of healing after endodontic surgery (meta-analysis):

Six meta-analyses of direct comparisons between GRT techniques (PL vs. control, MB vs. PL, MB + PL vs. MB, MB vs. control, Os vs. control, MB + Os vs. control) were carried out with the data obtained from the eleven selected RCTs. The meta-analysis of combined studies comparing MB-Os versus control (fixed effects model with the absence of heterogeneity; I^2^ = 0%) estimated a significant OR of 3.53 with a 95% confidence interval between 1.33 and 9.33. The remaining comparisons do not produce a significant OR (Figure 3).

Odds ratios were among the regeneration techniques for the success of healing after endodontic surgery (net meta-analysis).

Eleven RCTs (sixteen pairs of comparisons) were included in a frequentist NMA examining six GRT techniques (control, Os, PL, MB, MB + Os, and MB + PL) to analyze their influence on the success of healing after endodontic surgery. The data were combined with a fixed effects model (Mantel–Haenszel method). The nodes represent treatments, and the lines connecting the nodes are the six direct comparisons included in the NMA (Figure 4).

The outcome of GTR techniques was estimated in terms of OR and 95% confidence interval. OR > 1 indicated that the treatment in the first column on the left was superior to the comparator, while OR < 1 indicated the opposite. Statistically significant ORs are shown in bold (*p* < 0.05). Direct comparisons (6/15) are highlighted in gray, and indirect comparisons are uncolored. Only two statistically significant ORs were found (in bold) (*p* < 0.05). The probability of obtaining a successful result was 3.67 times greater in the MB + Os group than in the control group (*p* < 0.05). The success of healing was 3.47 times greater in the MB + Os group than in the MB group (*p* < 0.05). The remaining comparisons among the groups do not show significance (*p* > 0.05) (Table 3 and Figure 5).

The ranking of the GTR techniques was performed according to the P-score, which measures the degree of certainty and indicates whether one alternative is superior to the others. The P-score is measured on a scale of 0 to 1. The MB + Os group presents the highest P-score (0.93), followed by MB + PL (0.60) and PL (0.53) (Figure 6).

No heterogeneity or inconsistency was found in the NMA (test of heterogeneity/inconsistency Q = 0.29; *p* = 0.589). The net heat plot (Figure 7), which provides a detailed assessment of inconsistency, detected a very slight inconsistency between direct and indirect estimations, which was not significant.

### 3.5. Publication Bias

Six new studies were incorporated using the trim and fill method to adjust for funnel plot asymmetry, and a new OR for the six direct comparisons analyzed was estimated. No statistically significant differences were found with respect to the initially estimated OR (Figure 8).

## 4. Discussion

The objective of this systematic review and NMA was to investigate the influence of different GTR techniques used as adjuncts to endodontic surgery and analyze their efficacy, assessed in terms of success rates. The results of the NMA show that the success rate of endodontic surgery can be improved using GTR techniques as adjuncts, and combined therapy with bone grafts plus membranes results in a higher success rate.

Since the NMA did not show heterogeneity or inconsistency (Q = 1.16; *p* = 0.2821), the present NMA satisfied the assumption of transitivity, indicating that there were no systematic differences among the compared techniques other than the GTR techniques being compared [33]. Evaluating the transitivity assumption is critical, because the existence of intransitivity will bias treatment effect estimates [12]. Therefore, the calculated OR (3.6; *p* < 0.05) for the comparison between membrane plus bone grafting and endodontic surgery alone indicates that the success rate of this combination was almost four times higher than that of surgery without an adjunct GTR technique.

Most authors highlighted the relevance of membranes to promoting the healing of bone defects and preventing adjacent soft tissue ingrowth. The use of a membrane alone, without a bone graft, was 1.02 times more effective than endodontic surgery without a GTR technique (control), and more effective than platelet-rich plasma techniques. However, the membrane plus bone graft combination was 3.6 times more successful than membrane only.

The success rate of the combined membrane plus bone graft was 3.7 times higher than that of endodontic surgery alone (control). Parmar et al. (2019) reported a nonsignificant radiographic reduction in periapical bone defects regenerated using a resorbable collagen membrane. Complete periapical healing was observed in the control group, with rates of 60 to 80% and 53.3 to 73.3% of those of the membrane group, depending on the radiodiagnostic technique [32].

Marin-Botero et al. (2006) also reported that polyglactin-910 resorbable membranes had little influence on the complete healing of periapical bone defects after endodontic surgery (40%) compared with the control treatment (60%) [26]. Garret et al. (2002) reported that resorbable membranes did not show a statistically significant (*p* ˃ 0.05) radiographic reduction in periapical bone defects after endodontic surgery. Additionally, they did not recommend the use of resorbable membranes for bone defects with four walls that are confined to the apical region [34]. Santamaria Zuazua et al. (1998) analyzed the bone density and radiographic reduction in periapical bone defects after endodontic surgery using resorbable and non-resorbable membranes, and found no statistically significant difference (*p* ˃ 0.05) in bone density at 6 months after surgery between the two types of membranes.

These results suggest that GTR techniques using membranes do not contribute to increased periapical bone regeneration regardless of the membrane type [35]. However, Taschieri et al. (2011) retrospectively analyzed clinical and radiographic periapical bone healing after endodontic surgery procedures using a collagen resorbable membrane and recommended its application for through-and-through lesions [36]. Goyal et al. (2011) analyzed the impact of membranes and platelet-rich plasma on the complete periapical healing of periapical bone defects after endodontic surgery. They found no statistically significant differences (*p* ˃ 0.05) among the membrane alone, platelet-rich plasma alone, and the two combined [30]. Dhiman et al. (2015) reported no statistically significant difference (*p* ˃ 0.05) in the clinical and radiographic reduction in periapical bone defects after endodontic surgery using platelet-rich plasma techniques with respect to the control group [32].

Most authors reported that bone grafts stimulate bone defect healing and prevent adjacent soft tissue collapse [4,23,37,38,39]. Kattimani et al. (2014) highlighted the use of bovine-derived and synthetic hydroxyapatite bone grafts for the radiographic reduction in periapical bone defects after endodontic surgery. They found no statistically significant difference (*p* ˃ 0.05) in radiographic reduction between the two bone graft materials [38]. Kattimani et al. (2016) also compared the clinical and radiographic outcomes of bovine-derived and synthetic hydroxyapatite bone grafts after endodontic surgery. They found no statistically significant (*p* ˃ 0.05) difference between the two bone graft materials at 6-month follow-up [39]. Stassen et al. (1994) also analyzed the clinical and radiographic effects of bovine-derived hydroxyapatite bone grafts and did not recommend their use as adjuncts in endodontic surgery [23]. However, Sreedevi (2011) reported complete clinical and radiographic periapical bone healing after endodontic surgery using hydroxyapatite bone graft material with respect to the control group [4].

Other bone graft materials have been used as adjuncts to GTR techniques in endodontic surgery. Pantchev et al. (2009) retrospectively analyzed the clinical and radiographic outcomes of a synthetic bioactive glass material used as a bone graft after endodontic surgery. They found a statistically significant difference (*p* ˂ 0.05) at short-term follow-up (9–24 months), but no statistically significant difference (*p* ˃ 0.05) at long-term follow-up (33–48 months) [37]. It is more difficult to apply endodontic surgery using a GTR technique to 4-wall defects and through-and-through lesions because of the higher risk of soft tissue collapse and decreased stability of the bone regeneration material. Pecora et al. (2001) demonstrated that the addition of calcium sulfate as a bone graft material in GTR techniques for the treatment of through-and-through lesions improves the clinical outcome [24]. However, Taschieri et al. (2007, 2008) showed no statistically significant difference (*p* ˃ 0.05) after endodontic surgery when using resorbable collagen membrane and bovine-derived hydroxyapatite bone graft material for through-and-through lesions [27] and four-wall defects [29]. Tobon et al. (2002) reported that the simultaneous use of nonresorbable membrane and bovine-derived hydroxyapatite bone graft material produced complete clinical and radiographic periapical bone healing after endodontic surgery [25].

In addition, the wound healing scales and indices used in oral surgery do not capture the relationships between outcome parameters; therefore, Hamzani et al. (2018) proposed a novel scale that allows the assessment of wound healing phases [40]. Recently, Haj Yahya et al. (2020) described a novel procedure for measuring the healing process after surgical extraction based on an inflammatory proliferative remodeling scale that could also be used in further studies for the assessment of wound healing following endodontic surgery [41].

A limitation of this systematic review and meta-analysis is the possibility that not all articles related to the selection criteria were identified, although the risk was decreased because three databases were searched. In addition, most of the studies were of poor quality, according to the Cochrane Collaboration tool [16]. Furthermore, the most effective GTR technique (MB + Os) was only included in a single study. Therefore, further, better designed clinical studies with higher quality are necessary.

## 5. Conclusions

Within the limitations of this study, it was found that GTR techniques increased the success rate of endodontic surgery. The use of bone grafts plus membranes as an adjunct to surgical endodontic treatment promoted complete periapical bone healing, with a higher success rate, and improved the prognosis of endodontic surgery. Therefore, we recommend the use of bone grafts plus membranes as a GTR technique in endodontic surgery.

## Figures and Tables

**Figure 1 jcm-11-01062-f001:**
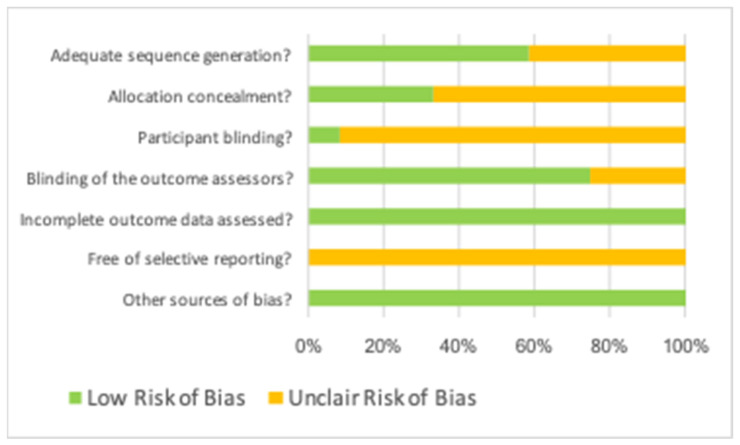
Risk of bias. Green color means “low risk of bias”, and yellow color means “unclair risk of bias”.

**Figure 2 jcm-11-01062-f002:**
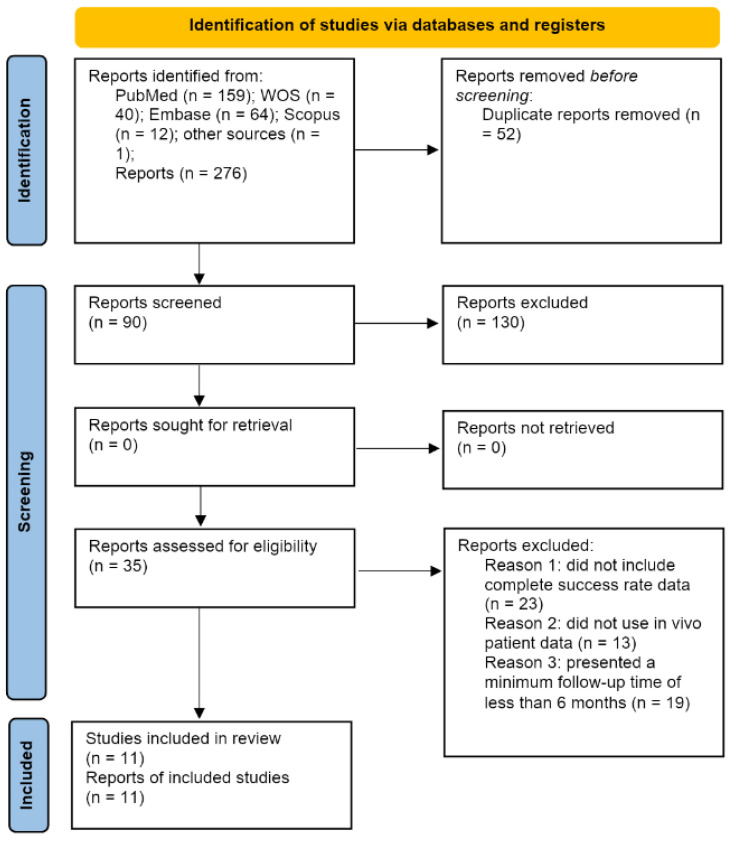
Preferred Reporting Items for Systematic Reviews and Meta-Analyses (PRISMA) flow diagram.

**Figure 3 jcm-11-01062-f003:**
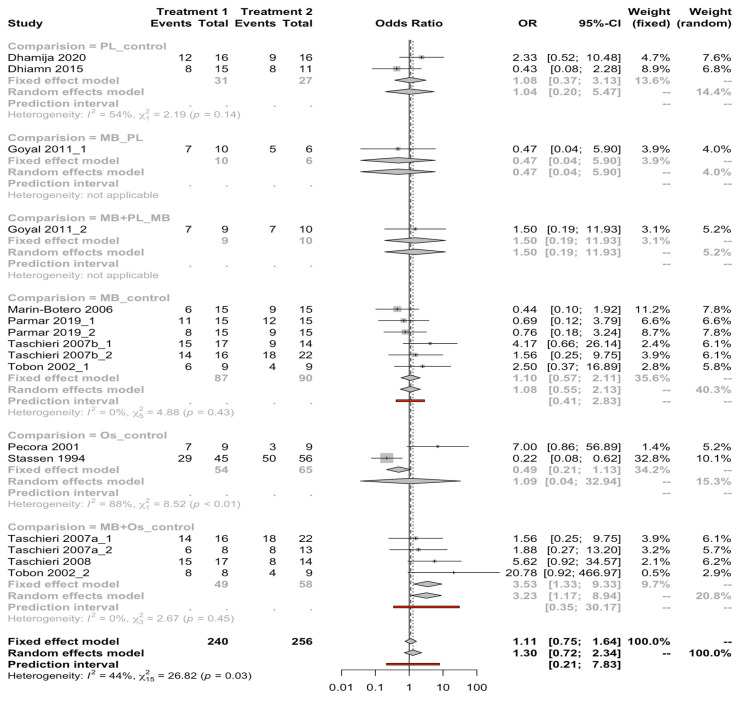
Forest plot of ORs among guided tissue regeneration techniques for healing success after endodontic surgery. Column 1 lists the articles included in the meta-analysis. Columns 2 and 3 show us the results of the articles in the form of a proportion. Column 3 is the forest plot itself, the graphic part of the representation. It plots the effect measures for each study on both sides of the null effect line, which is the one for the odds ratio. In the lower part of the graph, the global result of the meta-analysis is represented. Column 4 describes the estimated weight of each study in percentage, and column 5 presents the estimates of the weighted effect of each one. Diamonds indicate the mean and confidence interval of combined effect, and squares indicate the mean and confidence interval of each study. Red lines represent the prediction interval.

**Figure 4 jcm-11-01062-f004:**
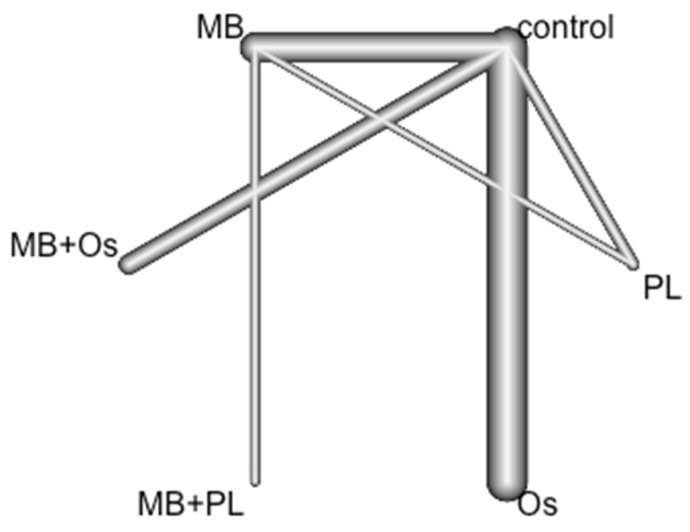
NETWORK plot of GTR techniques. Node size is proportional to the number of participants randomized to that technique, and the edge width is proportional to number of trials comparing two techniques.

**Figure 5 jcm-11-01062-f005:**
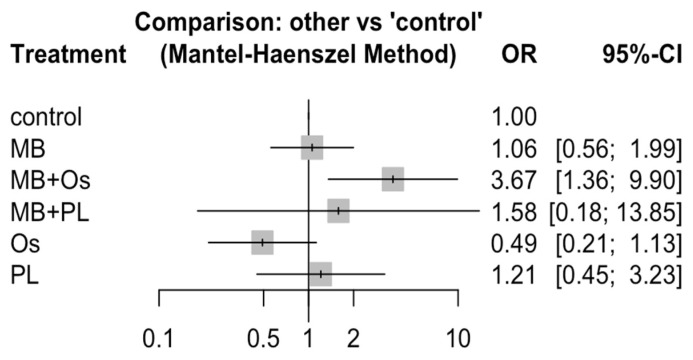
Forest plot of healing success using GTR techniques (odds ratio) compared to control group.

**Figure 6 jcm-11-01062-f006:**
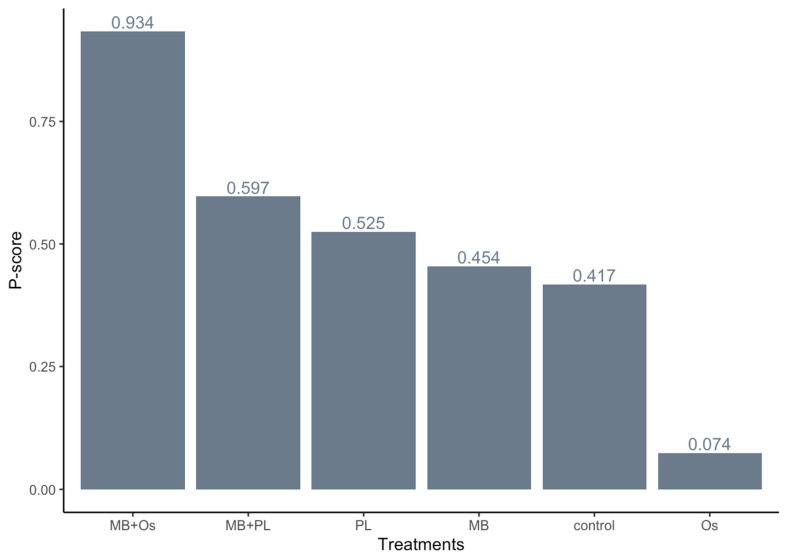
Ranking of GTR techniques by P-score.

**Figure 7 jcm-11-01062-f007:**
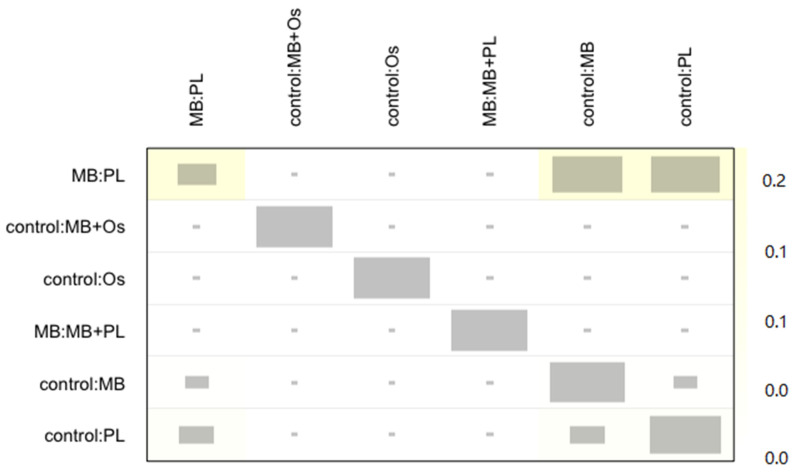
Net heat plot. Gray boxes signify the importance of one treatment comparison to the estimation of another treatment comparison. Larger boxes indicate more important comparisons. Color background, ranging from blue to red, signifies the inconsistency of comparison (row) attributable to design (column).

**Figure 8 jcm-11-01062-f008:**
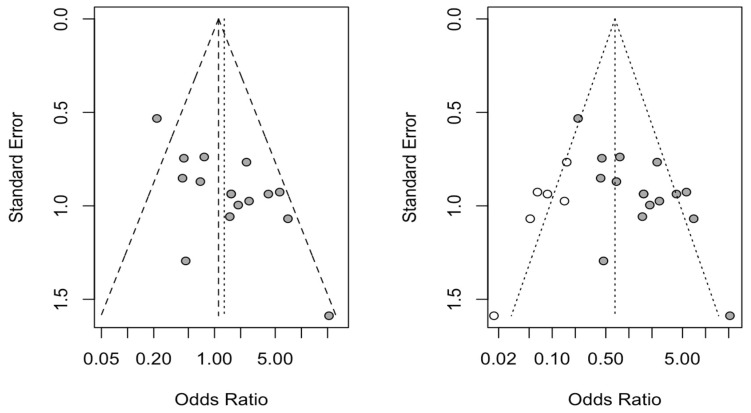
Initial funnel plot after trim and fill adjustment of OR of periapical healing among guided tissue regeneration techniques.

**Table 1 jcm-11-01062-t001:** Cochrane Collaboration tool for assessing risk of bias.

Author, Year	Adequate Sequence Generation?	Allocation Concealment?	Participant Blinding?	Blinding of Outcome Assessors?	Incomplete Outcome Data Assessed?	Free of Selective Reporting?	Other Sources of Bias?
Dhamija, 2020	Low	Low	Unclear	Low	Low	Unclear	Low
Dhiamn, 2015	Unclear	Low	Unclear	Low	Low	Unclear	Low
Goyal, 2011	Low	Low	Unclear	Low	Low	Unclear	Low
Marin Botero, 2006	Low	Low	Unclear	Low	Low	Unclear	Low
Parmar, 2019	Low	Unclear	Low	Low	Low	Unclear	Low
Pecora, 2002	Low	Low	Unclear	Low	Low	Unclear	Low
Stassen, 1994	Unclear	Unclear	Unclear	Low	Low	Unclear	Low
Taschieri, 2007a	Low	Unclear	Unclear	Low	Low	Unclear	Low
Taschieri, 2007b	Low	Unclear	Unclear	Low	Low	Unclear	Low
Taschieri, 2008	Low	Unclear	Unclear	Low	Low	Unclear	Low
Tobon, 2002	Unclear	Unclear	Unclear	Unclear	Low	Unclear	Low

**Table 2 jcm-11-01062-t002:** Qualitative analysis of articles included in systematic review.

Author/Year	Study Type	Sample (*n*)	Follow-Up Time (Months)	Measurement Procedure	GTR Technique	Complete Healing Rate	Periapical Healing Results
Dhiamn, 2015	RCT	26	12	Clinical and radiographic	Control	8/11	Control: 53.3% complete healing
					PL	8/15	PL: 53.33% complete healing
Goyal, 2011	RCT	25	3	Clinical and radiographic	MB	NAv	MB: 38.7 ± 22.3% periapical size reduction
					PL	NAv	PL: 39.2 ± 11.7% periapical size reduction
					PL + MB	NAv	PL + MB: 45.6 ± 14.2% periapical size reduction
		25	6		MB	NAv	MB: 67.9 ± 16.3% periapical size reduction
					PL	NAv	PL: 84.9 ± 10.4% periapical size reduction
					PL + MB	NAv	PL + MB: 75.9 ± 12.2% periapical size reduction
		25	9		MB	NAv	MB: 88.6 ± 10.1% periapical size reduction
					PL	NAv	PL: 93.3 ± 3.0% periapical size reduction
					PL + MB	NAv	PL + MB: 90.3 ± 6.9% periapical size reduction
		25	12		MB	7/10	MB: 97.0 ± 3.2 periapical size reduction
					PL	5/6	PL: 96.3 ± 3.0% periapical size reduction
					PL + MB	7/9	PL + MB: 97.3 ± 3.3% periapical size reduction
Marin Botero, 2006	RCT	30	12	Clinical and radiographic	Control	9/15	Control: 91.1 ± 18.1% periapical size reduction
					Mb	6/15	MB: 87.0 ± 18.6% periapical size reduction
					Os	50/68	
Parmar, 2019	RCT	30	12	Radiographic 2D	Control	12/15	Control: 12 ± 21mm^2^ (92 ± 12% reduction)
					MB	11/15	MB: 31 ± 30 mm^2^ (86 ± 14% reduction)
		30	12	Radiographic 3D	Control	9/15	Control: 174 ± 264 mm^3^ (85 ± 19% reduction)
					MB	8/15	MB: 324 ± 364 mm^3^ (82 ± 13% reduction)
Pecora, 2002	RCT	20	6	Clinical and radiographic	Control	3/10	Significant reduction in periapical defects (*p* ˂ 0.05)
					Os	8/10	
		18	12		Control	3/9	
					Os	7/9	
Dhamija, 2020	RTC	32	12	Clinical and radiographic	PL	9/16	Significant reduction in periapical defects (*p* ˂ 0.05)
					Control	5/16	
Stassen, 1994	RTC	101	24	Clinical and radiographic	Control	50/56	No significant reduction in periapical defects (*p* = 0.057)
					Os	29/45	
Taschieri, 2007	RTC	59	12	Radiographic 4-wall defects	Control	18/22	Control: 80.0–83.3% complete healing
					MB + Os	14/16	MB + Os: 81.8–100% complete healing
		59	12	Radiographic through-and-through	Control	8/13	Control: 55.6–75.0% complete healing
					MB + Os	6/8	MB + Os: 75.0% complete healing
Taschieri, 2008	RTC	31	12	Clinical and radiographic	Control	8/14	Control: 57.1% complete healing
					MB + Os	15/17	Os: 88.2% complete healing
Taschieri, 2008	RTC	69	12	Clinical and radiographic 2-wall defects	Control	9/14	Statistically significant differences (*p* = 0.02)
					MB + Os	15/17	
				Clinical and radiographic 4-wall defects	Control	18/22	No statistically significant differences (*p* = 0.21).
					MB + Os	14/16	
Tobon, 2002	RTC	26	12	Radiographic	Control	4/9	Control: 44.4% complete healing
					MB	6/9	MB: 66.6% complete healing
					MB + Os	8/8	MB + Os: 100% complete healing

RCT, randomized controlled trial; CT, controlled trial; CS, case series; NAv, not available; PL, platelet enriched plasma; Os, bone graft; MB, membrane.

**Table 3 jcm-11-01062-t003:** Comparison between GTR techniques using OR and 95% confidence intervals estimated in Netmeta. * *p* < 0.05.

	Control	MB	MB + Os	MB + PL	Os	PL
Control	1	0.95 0.50; 1.78	0.27 * 0.10; 0.73	0.63 0.07; 5.52	2.04 0.88; 4.66	0.82 0.31; 2.21
MB	1.06 0.56; 1.99	1	0.29 * 0.09; 0.94	0.66 0.08; 5.30	2.14 0.76; 6.11	0.87 0.29; 2.68
MB + Os	3.67 * 1.36; 9.90	3.47 * 1.07; 11.3	1	2.31 0.21; 25.1	7.46 2.04; 27.2	3.04 0.75; 12.3
MB + PL	1.58 0.18; 13.9	1.50 0.19; 11.9	0.43 0.04; 4.69	1	3.22 0.32; 32.9	1.31 0.12; 13.8
Os	0.49 0.21; 1.13	0.47 0.16; 1.32	0.13 0.04; 0.49	0.31 0.03; 3.16	1	0.41 0.11; 1.47
PL	1.21 0.45; 3.23	1.14 0.37; 3.49	0.33 0.08; 1.33	0.76 0.07; 8.04	2.46 0.68; 8.32	1

## Data Availability

Information is available upon request in accordance with relevant restrictions (e.g., privacy or ethical).

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
