# Peer review of "Influence of Guided Tissue Regeneration Techniques on the Success Rate of Healing of Surgical Endodontic Treatment: A Systematic Review and Network Meta-Analysis"

_jcm, 2022, doi:10.3390/jcm11041062_

Round 1
Reviewer 1 Report
There are some weaknesses through the manuscript which need improvement. Therefore, the submitted manuscript cannot be accepted for publication in this form, but it has a chance of acceptance after a minor revision. My comments and suggestions are as follows:
1- Abstract gives information on the main feature of the performed study, but a couple of sentences about the background of the study must be added.
2- Authors must clarify necessity of the performed research. Objectives of the study must be clearly mentioned in introduction.
3- The literature study must be enriched. In this respect, authors must read and refer to the following papers: (a) https://doi.org/10.1186/s12859-020-3351-y (b) https://doi.org/10.1111/idj.12622 and other research works.
4- It would be nice, if authors could add some figures form previous research works.
5- The main reference of each formula must be cited. Moreover, each parameters in equations must be introduced. Please double check this issue.
6- All figures must be illustrated in a high quality.
7- Standard deviation is the presented curves must be discussed
8- In its language layer, the manuscript should be considered for English language editing. There are sentences which have to be rewritten.
9- The conclusion must be more than just a summary of the manuscript. List of references must be updated based on the proposed papers. Please provide all changes by red color in the revised version.
Author Response
Dear Reviewer 1,
I’m pleased to resubmit the manuscript of the work entitled, “Influence of Several Guided Tissue Regeneration Techniques on the Success Rate of Healing of the Surgical Endodontic Treatment: Systematic Review and Network Meta-Analysis”
Reviewer 1: English language and style are fine/minor spell check required
Response: In order to adapt to the reviewer's 1 comments, we have sent the manuscript to the English Editing Service of MDPI. We attached the Certificate.
Reviewer 1: Abstract gives information on the main feature of the performed study, but a couple of sentences about the background of the study must be added
Response: In order to adapt to the reviewer's 1 comments, we have added a sentence with the background related to the study.
Reviewer 1: Authors must clarify necessity of the performed research. Objectives of the study must be clearly mentioned in introduction
Response: In order to adapt to the reviewer's 1 comments, we clarify that the necessity of the performed research is described in the fifth sentence of the Introduction section: “Numerous studies have reported the clinical effectiveness of GTR techniques to promote healing and improve the outcome of surgical endodontic treatments [10,11]. However, the wide range of biomaterials available, the GTR treatment protocols and the lack of standardization in assessment criteria generate inconsistent and confusing results. Therefore, an evidence-based review and meta-analysis of the available literature regarding the influence of GTR techniques on the outcome of surgical endodontic treatment is necessary”. Moreover, we have added a sentence to better clarify the necessity of the performed research, following the recommendations of the Reviewer 1.
Reviewer 1: The literature study must be enriched. In this respect, authors must read and refer to the following papers: (a) https://doi.org/10.1186/s12859-020-3351-y (b) https://doi.org/10.1111/idj.12622 and other research works
Response: In order to adapt to the reviewer's 1 comments, we have added a paragraph and references in the Discussion section: “In addition, the wound healing scales and indexes used in oral surgery prevent the relationship between the outcome parameters; therefore, Hamzani et al. (2018) proposed a novel scale that allow the assessment among the wound healing phases [Hamzani Y, Chaushu G. Evaluation of early wound healing scales/indexes in oral surgery: A literature review. Clin Implant Dent Relat Res. 2018, 20, 1030-1035. doi: 10.1111/cid.12680]. Recently, Haj Yahya et al. (2020) described a novel measurement procedure of the healing process after the surgical extraction based on an inflammatory proliferative remodeling scale that could be also used in further studies for the assessment of the wound healing following endodontic surgery [Haj Yahya B, Chaushu G, Hamzani Y. Evaluation of wound healing following surgical extractions using the IPR Scale. Int Dent J. 2020. doi: 10.1111/idj.12622. Epub ahead of print]”. We apologized because we did not find the reference “https://doi.org/10.1186/s12859-020-3351-y”.
Reviewer 1: It would be nice, if authors could add some figures form previous research works
Response: In order to adapt to the reviewer's 1 comments, we apologize because we do not have figures from previous research works; however, we will conduct related research works soon.
Reviewer 1: The main reference of each formula must be cited. Moreover, each parameter in equations must be introduced. Please double check this issue
Response: In order to adapt to the reviewer's 1 comments, we have revised the full document and we have confirmed that all the formulas are cited, except for the Q test. We have included the following reference to document the Q test: Dias S, Welton NJ, Caldwell DM, Ades AE. Checking consistency in mixed treatment comparison meta-analysis. Stat Med. 2010 Mar 30;29(7-8):932-44. doi: 10.1002/sim.3767.
Reviewer 1: All figures must be illustrated in a high quality
Response: In order to adapt to the reviewer's 1 comments, we have improved the quality of Figure 2 and 7.
Reviewer 1: In its language layer, the manuscript should be considered for English language editing. There are sentences which have to be rewritten
Response: In order to adapt to the reviewer's 1 comments, we have sent the manuscript to the English Editing Service of MDPI. We attached the Certificate.
Reviewer 1: The conclusion must be more than just a summary of the manuscript. List of references must be updated based on the proposed papers. Please provide all changes by red color in the revised version
Response: In order to adapt to the reviewer's 1 comments, we have added a final recommendation in the Results section. In addition, we have also updated the reference list with 5 new references; including the proposed papers.
We take this opportunity to thank the recommendations and suggestions made by the reviewers to improve the document.
Reviewer 2 Report
The study is interesting, and above all it was conducted with correct methodology. some aspects of the systematic review must however be clarified and some considerations on the network must be made.
- Line 152: Does I2 refer to the higgins index? Specify it.
- line 156: specify the figure of the forest plot;
- line 177: specify the type of gray literature (opengrey, google sholar abstract and conference poster) with the related records identified.
- Figure 2: I would use the 2020 flow chart prism.
- provide more description in figure 3
- Indicate how to remove duplicates (manually or by program for example ENDnote)
- For the sake of completeness, I would also have investigated the bibliography of previous reviews on the subject, if it has been performed and not reported it can be added to other sources)
- In the qualitative analysis, also briefly add general information such as the total number of patients (including all studies) divided perhaps by the different techniques.
- In the results section I would also add comments with the main conclusions and results of each study included.
- add in the discussion how these assumptions have been investigated: similarity, transitivity and coherence
- Tobon, 2002 [Error! Bookmark not defined.]?
Author Response
Dear Reviewer 2,
I’m pleased to resubmit the manuscript of the work entitled, “Influence of Several Guided Tissue Regeneration Techniques on the Success Rate of Healing of the Surgical Endodontic Treatment: Systematic Review and Network Meta-Analysis”
Reviewer 2: I don't feel qualified to judge about the English language and style
Response: In order to adapt to the reviewer's 2 comments, we have sent the manuscript to the English Editing Service of MDPI. We attached the Certificate.
Reviewer 2: Line 152: Does I2 refer to the higgins index? Specify it
Response: In order to adapt to the reviewer's 2 comments, we confirm that the I² statistic describes the percentage of variation across studies that is due to heterogeneity rather than chance. I² = 100% x (Q-df)/Q. I² is an intuitive and simple expression of the inconsistency of studies’ results. Additionally, we have added a reference: Higgins JP, Thompson SG, Deeks JJ, Altman DG. Measuring inconsistency in meta-analyses. BMJ. 2003 Sep 6;327(7414):557-60. doi: 10.1136/bmj.327.7414.557.
Reviewer 2: line 156: specify the figure of the forest plot;
Response: In order to adapt to the reviewer's 2 comments, we have specified the figure of the Forest Plot adding the following sentence at the Material and Methods section: “Column 1 lists the articles included in the meta-analysis. Columns 2 and 3 show us the results of the articles in the form of a proportion. Column 3 is the forest plot itself, the graphic part of the representation. It plots the effect measures for each study on both sides of the null effect line, which is the one for odds ratio. In the lower part of the graph, the global result of the meta-analysis is represented. Column 4 describes the estimate of the weight of each study in percentages and a fifth column with the estimates of the weighted effect of each one”.
Reviewer 2: line 177: specify the type of gray literature (opengrey, google sholar abstract and conference poster) with the related records identified
Response: In order to adapt to the reviewer's 2 comments, we clarified in the Results section that the article found in gray literature found in the bibliography of a previous review (Liu TJ, Zhou JN, Guo LH. Impact of different regenerative techniques and materials on the healing outcome of endodontic surgery: a systematic review and meta-analysis. Int Endod J. 2021 Apr;54(4):536-555. doi: 10.1111/iej.13440.).
Reviewer 2: Figure 2: I would use the 2020 flow chart prism
Response: In order to adapt to the reviewer's 2 comments, we have replaced Figure 2 with the 2020 flowchart prism.
Reviewer 2: provide more description in figure 3
Response: In order to adapt to the reviewer's 2 comments, we have provided more description in Figure 3. And red lines represent the prediction interval
Reviewer 2: Indicate how to remove duplicates (manually or by program for example ENDnote)
Response: In order to adapt to the reviewer's 2 comments, we clarify that duplicates were removed using RefWorks (https://refworks.proquest.com/reference/upload/recent/).
Reviewer 2: For the sake of completeness, I would also have investigated the bibliography of previous reviews on the subject, if it has been performed and not reported it can be added to other sources)
Response: In order to adapt to the reviewer's 2 comments, we clarified in the Results section that the article found in gray literature found in the bibliography of a previous review (Liu TJ, Zhou JN, Guo LH. Impact of different regenerative techniques and materials on the healing outcome of endodontic surgery: a systematic review and meta-analysis. Int Endod J. 2021 Apr;54(4):536-555. doi: 10.1111/iej.13440.).
Reviewer 2: In the qualitative analysis, also briefly add general information such as the total number of patients (including all studies) divided perhaps by the different techniques.
Response: In order to adapt to the reviewer's 2 comments, we clarify that this is already performed in the Table 2, even it is necessary to perform the meta-analysis.
Reviewer 2: In the results section I would also add comments with the main conclusions and results of each study included
Response: In order to adapt to the reviewer's 2 comments, we clarify that this is already performed in the Table 2
Reviewer 2: add in the discussion how these assumptions have been investigated: similarity, transitivity and coherence
Response: In order to adapt to the reviewer's 2 comments, we clarify that assuming transitivity implies that the selection criteria used in the systematic review have shown that the groups studied in the different articles are comparable in aspects such as study design, the populations studied, duration of treatment, the definition of the outcome variable analyzed or the distribution of factors that can affect or modify the effect of the different treatments. Using the Q test for heterogeneity/inconsistency, it is assessed statistically and we also incorporate the most detailed analysis of the inconsistency between the dirty and indirect comparisons with the netheat plot.
Reviewer 2: Tobon, 2002 [Error! Bookmark not defined.]?
Response: In order to adapt to the reviewer's 2 comments, we have removed the typo error.
We take this opportunity to thank the recommendations and suggestions made by the reviewers to improve the document.
Round 2
Reviewer 2 Report
The authors answered all the questions posed on the net work meta-analysis, however I believe that there are still some considerations to be made;
minor revisions:
Tthe following sentence should perhaps be placed below the figure 3 as an explanation, perhaps my indications have been unclear
(Column 1 lists the articles included in the metaanalysis. Columns 2 and 3 show us the results of the articles in the form of a proportion. Column 3 is the forest plot itself, the graphic part of the representation. It plots the effect measures for each study on both sides of the null effect line, which is the one for odds ratio. In the lower part of the graph, the global result of the meta-analysis is represented. Column 4 describes the estimated of the weight of each study in percentage, s and a fifth column 5 gives with the estimates of the weighted effect of each one)
Author Response
Dear Reviewer 2:
I’m pleased to resubmit the manuscript of the work entitled, “Influence of Several Guided Tissue Regeneration Techniques on the Success Rate of Healing of the Surgical Endodontic Treatment: Systematic Review and Network Meta-Analysis”
Reviewer 2: The following sentence should perhaps be placed below the figure 3 as an explanation, perhaps my indications have been unclear
(Column 1 lists the articles included in the metaanalysis. Columns 2 and 3 show us the results of the articles in the form of a proportion. Column 3 is the forest plot itself, the graphic part of the representation. It plots the effect measures for each study on both sides of the null effect line, which is the one for odds ratio. In the lower part of the graph, the global result of the meta-analysis is represented. Column 4 describes the estimated of the weight of each study in percentage, s and a fifth column 5 gives with the estimates of the weighted effect of each one)
Response: In order to adapt to the reviewer's 2 comments, we have placed the sentence below the Figure 3.
We take this opportunity to thank the recommendations and suggestions made by the reviewers to improve the document.
Yours sincerely,